# Rapid Development of Modified Vaccinia Virus Ankara (MVA)-Based Vaccine Candidates Against Marburg Virus Suitable for Clinical Use in Humans

**DOI:** 10.3390/vaccines12121316

**Published:** 2024-11-24

**Authors:** Alina Tscherne, Georgia Kalodimou, Alexandra Kupke, Cornelius Rohde, Astrid Freudenstein, Sylvia Jany, Satendra Kumar, Gerd Sutter, Verena Krähling, Stephan Becker, Asisa Volz

**Affiliations:** 1Division of Virology, Department of Veterinary Sciences, Ludwig Maximilians University (LMU Munich), 85764 Oberschleißheim, Germany; a.tscherne@lmu.de (A.T.); georgia.kalodimou@viro.vetmed.uni-muenchen.de (G.K.); astrid.freudenstein@viro.vetmed.uni-muenchen.de (A.F.); sylvia.jany@viro.vetmed.uni-muenchen.de (S.J.); satyendra.kumar@lmu.de (S.K.); gerd.sutter@lmu.de (G.S.); 2German Center for Infection Research, Partner Site Munich, 85764 Oberschleißheim, Germany; 3Institute of Virology, Philipps University Marburg, 35037 Marburg, Germany; kupke@staff.uni-marburg.de (A.K.); rohdecor@staff.uni-marburg.de (C.R.); becker@staff.uni-marburg.de (S.B.); 4German Center for Infection Research, Partner Site Gießen-Marburg-Langen, 35037 Marburg, Germany; 5Institute of Virology, University of Veterinary Medicine Hannover, 30559 Hannover, Germany; 6German Center for Infection Research, Partner Site Hannover-Braunschweig, 30559 Hannover, Germany

**Keywords:** Modified Vaccinia virus Ankara (MVA), viral vector vaccine, emerging viruses, rapid vaccine development, Marburg virus

## Abstract

Background/Objectives: Marburg virus (MARV) is the etiological agent of Marburg Virus Disease (MVD), a rare but severe hemorrhagic fever disease with high case fatality rates in humans. Smaller outbreaks have frequently been reported in countries in Africa over the last few years, and confirmed human cases outside Africa are, so far, exclusively imported by returning travelers. Over the previous years, MARV has also spread to non-endemic African countries, demonstrating its potential to cause epidemics. Although MARV-specific vaccines are evaluated in preclinical and clinical research, none have been approved for human use. Modified Vaccinia virus Ankara (MVA), a well-established viral vector used to generate vaccines against emerging pathogens, can deliver multiple antigens and has a remarkable clinical safety and immunogenicity record, further supporting its evaluation as a vaccine against MARV. The rapid availability of safe and effective MVA-MARV vaccine candidates would expand the possibilities of multi-factored intervention strategies in endemic countries. Methods: We have used an optimized methodology to rapidly generate and characterize recombinant MVA candidate vaccines that meet the quality requirements to proceed to human clinical trials. As a proof-of-concept for the optimized methodology, we generated two recombinant MVAs that deliver either the MARV glycoprotein (MVA-MARV-GP) or the MARV nucleoprotein (MVA-MARV-NP). Results: Infections of human cell cultures with recombinant MVA-MARV-GP and MVA-MARV-NP confirmed the efficient synthesis of MARV-GP and MARV-NP proteins in mammalian cells, which are non-permissive for MVA replication. Prime-boost immunizations in C57BL/6J mice readily induced circulating serum antibodies binding to recombinant MARV-GP and MARV-NP proteins. Moreover, the MVA-MARV-candidate vaccines elicited MARV-specific T-cell responses in C57BL/6J mice. Conclusions: We confirmed the suitability of our two backbone viruses MVA-mCherry and MVA-GFP in a proof-of-concept study to rapidly generate candidate vaccines against MARV. However, further studies are warranted to characterize the protective efficacy of these recombinant MVA-MARV vaccines in other preclinical models and to evaluate them as vaccine candidates in humans.

## 1. Introduction

Marburg virus (MARV), the causative agent of the rare but severe hemorrhagic Marburg Virus Disease (MVD), is a significant threat to human and non-human primates. An infection with MARV results in a severe hemorrhagic fever disease with organ involvement, including spleen, liver, renal tissue, and coagulation dysfunction. Other reported symptoms of MVD include bradycardia, conjunctivitis, nausea, severe watery diarrhea, vomiting, muscle pains, aggression, and confusion [1,2,3,4]. The first human cases were reported in 1967 [5] in Marburg, Frankfurt am Main (Germany), and Belgrade (Serbia; former Yugoslavia). Since then, smaller outbreaks have occurred periodically in Africa [3,6], with a case fatality rate of up to 90%. The two largest outbreaks of MARV occurred in 1998–2000 in the Democratic Republic of Congo (DRC) and 2005 in Angola [1,3,7], with more than 152 and 374 reported cases, respectively. The case fatality rates ranged between 83% (DRC) and 88% (Angola) [1]. Between 2007 and 2023, only a few cases (between 1 and 15 cases) have been reported annually in Africa, also including countries where the virus has not been detected before, such as Guinea (2021), Ghana (2022), and Equatorial Guinea (2023) [3]. Furthermore, MARV was detected in 2007 and 2012 in Uganda, followed by a re-emergence of the virus in 2014 and 2017 [1]. In addition, Rwanda reported its first cases of MVD by 27 September 2024, and until now, more than 61 cases with more than 14 deaths [CFR: 23%] have been confirmed [8]. Thus, the current circulation of MARV in Rwanda is the third-largest outbreak of MARV since its first emergence in 1967. Due to the increased MVD cases and the spread into non-endemic countries over recent years, the WHO has recognized MARV as a significant pathogen capable of efficiently spreading at a national level, highlighting the need for effective vaccines and treatments [3]. Due to the complex epidemiological situation in the African continent, an availability of different vaccine candidates such as mRNA-, Adenovirus-, and MVA-based vaccination strategies that are suited for use in different target populations is highly warranted [9].

MARV belongs to the order *Mononegavirales,* the family Filoviridae, and the genus *Orthomarburgvirus* [10,11]. Orthomarburgviruses are enveloped, single-stranded, non-segmented RNA viruses of negative polarity, and they encode for seven different proteins [12,13,14]: glycoprotein (GP), nucleoprotein (NP), virion protein (VP) 24, VP30, VP35, VP40, and the viral polymerase L. The glycoprotein is a trimer on the viral surface and is responsible for the attachment to and entry of cells [15,16,17]. Each monomer comprises two subunits, GP1 (receptor-binding core) and GP2 (two heptad repeats, transmembrane domain), connected by a disulfide bond. The nucleoprotein encapsulates the viral RNA and forms the nucleocapsid complex, which is important for the transcription and translation of viral RNA [18,19].

The rapid availability of safe and effective vaccines is an urgent need in an outbreak scenario. This was vividly demonstrated during the COVID-19 pandemic, when different vaccines rapidly received emergency authorization for use in humans. Upon these, mRNA-based COVID-19 vaccines were one of the first vaccines to be available for human use [for review see [20]]. Due to their rapid construction, mRNA vaccines present a safe first line of defense to combat newly emerging pathogens [21,22]. However, mRNA vaccines require low-temperature storage [22], which limits their application in areas with poor conditions and low economic levels. Due to this, additional vaccination strategies need to be considered for MARV vaccine development [23]. Viral vectors also represent another promising approach for generating safe and effective vaccines suitable for use against emerging viruses. Recombinant poxviruses are well-established viral vector tools and have been used for decades to develop novel vaccines with prophylactic or therapeutic applications against (re)-emerging infections [24,25,26]. Key features that contributed to the success of poxviral vectors as a vaccine platform include the capacity to insert large heterologous DNA sequences, viral-specific control of recombinant gene expression, and their high immunogenicity and efficacy by inducing both cellular immune responses and antigen-specific antibodies. Furthermore, poxviral vectors are highly robust and genetically stable, and virus production under Good Manufacturing Practice (GMP) for amplification at an industrial scale is easy to implement.

Among these recombinant poxviruses, one promising and well-established candidate is Modified Vaccinia virus Ankara (MVA) [24,27,28]. MVA, a highly attenuated member of the genus *Orthopoxvirus*, was generated by passing vaccinia virus (VACV) more than 500 times on chicken embryonic fibroblast (CEF) cell cultures [29]. As a result, MVA lost > 14% of genetic information and, thereby, the ability to fulfill its complete replication cycle to release infectious particles in mammalian cells, thus contributing to the high safety profile of MVA [30]. MVA-based candidate vaccines are manufactured by infecting avian cells (e.g., CEF cells or DF-1 cells), in which MVA can still fulfill its replication cycle, allowing for the generation of viral stocks with high viral titers. Manufacturing companies continuously improve their protocols to speed up the manufacturing process, e.g., using permanent cell lines instead of primary CEF cells [31,32]. 

Over the last few years, several promising MVA-based candidate vaccines have been generated for use in humans and have been evaluated successfully for immunogenicity in various preclinical [33,34,35,36,37,38] and clinical studies [39,40,41,42]. Particularly during the COVID-19 pandemic, several MVA-based candidate vaccines against SARS-CoV-2 [35,36,37,38,43,44,45,46,47,48] had been designed, and several of them were approved for evaluation in clinical studies (see ClinicalTrials.gov: NCT05226390, NCT05950776, NCT04895449; NCT04569383; NCT04977024) rapidly due to the well-known safety and highly immunogenic profile of MVA. However, the generation of recombinant MVA vector viruses is more time-consuming than mRNA vaccine generation and can take 25–49 weeks. This highlights the need to optimize the generation process of MVA-based vaccines further.

Different from mRNA vaccines, the generation process of MVA-based vaccines includes the insertion of the target antigen into the backbone virus, the isolation of a positive clonal isolate, the propagation and purification of the virus, and in vitro characterization to confirm genetic stability and unimpaired expression of the target gene. We have established standardized quality control procedures [49], allowing for swift testing of newly generated MVA-based candidate vaccines under pre-GMP conditions. However, the isolation of a clonal isolate of a recombinant MVA is the most crucial part, but also the most laborious one; therefore, we aimed to improve our MVA vector technology for a facilitated and more rapid isolation of recombinant MVAs for vaccine development against emerging and re-emerging viruses.

Using the methodology of homologous recombination to insert foreign gene sequences into the MVA-genome by a specific transfer plasmid, there are several screening methods to discriminate between parental and recombinant MVAs, using either fluorescent marker genes [49,50,51,52] (e.g., GFP, mCherry) or selection markers [53,54,55,56,57,58,59,60], such as xanthine-guanine phosphoribosyltransferase, thymidine kinase, β-galactosidase, or β-glucuronidase. We recently established a protocol for generating MVA-based vaccines using a single fluorescent marker to isolate recombinant MVAs that meet the requirements for human clinical applications [49]. To more swiftly isolate clonal, marker-free recombinant MVA viruses, we further generated novel MVA-backbone viruses, sfMVA-GFP and sfMVA-mCherry, allowing for the generation of recombinant MVA viruses suitable for production under GMP conditions through simple color screening, following homologous recombination between the viral genome and foreign DNA by the combined use of different fluorescent proteins such as green fluorescent protein (GFP) or red fluorescent protein (mCherry).

Using these newly established sfMVA-GFP and sfMVA-mCherry backbone viruses as proof of principle, we generated two recombinant MVA viruses expressing either the glycoprotein (MVA-MARV-GP) or the nucleoprotein (MVA-MARV-NP) of the re-emerging MARV. We performed extensive in vitro and in vivo characterization. MVA-MARV-NP and MVA-MARV-GP demonstrated genetic identity and stability. They induced robust MARV-specific antibody titers and T cell activation in mice using a prime-boost immunization strategy, hereby confirming the suitability of the two MVA backbone viruses for the rapid generation of novel vaccines against new emerging viruses.

## 2. Materials and Methods

### 2.1. Cell Cultures

Primary chicken embryo fibroblasts (CEF) were isolated under serum-free conditions following established protocols [49]. DF-1 cells (ATCC^®^ CRL-12203™, ATCC, Manassas, VA, USA), Vero cells (ATCC CCL-81, ATCC, Manassas, VA, USA), HeLa cells (ATCC CCL-2, ATCC, Manassas, USA), HaCat cells (CLS Cell Lines Service GmbH, Eppelheim, Germany), and A549 cells (ATCC^®^ CCL-185™, ATCC, Manassas, VA, USA) were cultured as published elsewhere [36,49] and kept at 37 °C and 5% CO_2_. 

### 2.2. Construction of Plasmids pLW73-mCherry and pG06-GFP and Generation of Recombinant MVA Backbone Viruses

The MVA transfer plasmids pLW73-mCherry and pG06-GFP were designed and subsequently used to generate the backbone viruses MVA-mCherry and MVA-GFP, respectively, following established protocols as published elsewhere [36,49]. The codon-optimized sequences of full-length mCherry and GFP (green fluorescent protein) (synthesized by Eurofins, Ebersberg, Germany) were introduced into the MVA transfer plasmids pLW73 and pG06, respectively, and the expression of the proteins was placed under transcriptional control of either the vaccinia virus (VACV)-specific late promoter PmH5 [61] or p11.

### 2.3. Construction of pIII-MARV-GP and pLW73-MARV-NP and Generation of Recombinant MVA Vaccine Candidates

The MVA transfer plasmids pIII-MARV-GP (pIII-GP) and pLW73-MARV-NP (pLW73-NP) were designed and subsequently used to generate the vaccine candidates MVA-MARV-GP (MVA-GP) and MVA-MARV-NP (MVA-NP), respectively, following established protocols as published elsewhere [36,49]. The full-length coding sequences of the MARV Guinea strain [62] GP (Genbank ID: OK665848) and the NP (Genbank ID: OK665848) were codon-optimized for improved expression by MVA. In addition, since information on the first nucleotides of the NP was not available, the missing nucleotides were refilled with those from the Angola strain (Genbank ID: DQ447654). The modified cDNAs were produced by DNA synthesis (Eurofins, Ebersberg, Germany) and cloned into the MVA transfer plasmids pIIIH5red or pLW73 [63] to obtain the MVA transfer plasmids pIIIH5red-MARV-GP (pIIIH5red-GP) and pLW73-MARV-NP (pLW73-NP), respectively. The expressions of recombinant MARV-GP and MARV-NP were placed under transcriptional control of the VACV-specific early/late PmH5 promoter [61].

### 2.4. In Vitro Characterization of Recombinant MVA Backbone Viruses and MVA Vaccine Candidates

MVA viruses MVA-mCherry, MVA-GFP, MVA-MARV-NP, and MVA-MARV-GP were analyzed for genetic stability using polymerase chain reaction (PCR) and for replicative capacity using multiple-step growth experiments, following well-established and standardized protocols as published elsewhere [36,49]. To confirm genetic stability, oligonucleotide sequences targeting the six major deletion sites (Del I–VI) and C7L gene locus were used (MVA Del1-F: CTTTCGCAGCATAAGTAGTATGTC, MVA_Del1-R: CATTACCGCTTCATTCTTATATTC; MVA_Del2-F: GGGTAAAATTGTAGCATCATATACC MVA_Del2-R: AAAGCTTTCTCTCTAGCAAAGATG; MVA_Del3-F: GATGAGTGTAGATGCTGTTATTTTG; MVA_Del3-R: GCAGCTAAAAGAATAATGGAATTG; MVA_Del4-F: AGATAGTGGAAGATACAACTGTTACG; MVA_Del4-R: TCTCTATCGGTGAGATACAAATACC; MVA_Del5-F: CGTGTATAACATCTTTGATAGAATCAG; MVA_Del5-R: AACATAGCGGTGTACTAATTGATTT, MVA_Del6-F: CTACAGGTTCTGGTTCTTTATCCT; MVA_Del6-R: CACGGTCAATTAACTATAGCTCTG; C7L-F: CATGGACTCATAATCTCTATAC, C7L-R: ATGGGTATACAGCACGAATTC). Analysis of the correct insertion of the target antigen into the region between the two MVA genes, MVA-069R and MVA-070L, was conducted using the primers MVA-069R/070L-5′: ATTCTCGCTGAGGAGTTGG and MVA-069R/070L-3′: GTCGTGTCTACAAAAGGAG [49]. The genetic identity of MARV-GP and MARV-NP was confirmed by using primers targeting the inserted sequence: MARV-NP F1: CTCTTCCTCCCAAAACTTGTC, MARV-NP R1: CTTCCTTTCCTCGTCATCTTC; MARV-NP F2: GAAGATGACGAGGAAAGGAAG, MARV-NP R2: TGAGCATACAACGGAGGAG; MARV-GP F1: TCAATACCACAGACACAAACAG, MARV-GP R1: TCCACCATTTACCACCCAG; MARV-GP F2: ACAACAATGTATCGAGGCAAAG, MARV-GP R2: CGAGTTAGGTGTAGGAGAGG.

### 2.5. Western Blot Analysis of Recombinant Proteins

Unimpaired and correct expression of recombinant mCherry, GFP, MARV-GP, and MARV-NP were tested by infecting monolayers of HeLa, A549, VeroE6, DF-1, or CEF cells with recombinant MVA-mCherry, MVA-GFP, MVA-GP, or MVA-NP at MOI 5 and subsequent Western blot analysis. At different time points (0, 8, 24, 48 h post infection (hpi)), cell lysates were prepared and additionally, in the cases of MARV-GP, deglycosylation was conducted by treating the cell lysates with PNGase F (New England Biolabs, Frankfurt am Main, Germany). The conditions for Western blot have already been published elsewhere [36,49]. The expression of recombinant proteins was confirmed by using primary antibodies targeting GFP, mCherry, MARV-NP, or MARV-GP (Appendix A). For loading control, a rabbit anti-GAPDH antibody was used (Appendix A). Anti-mouse or anti-rabbit IgG antibodies, conjugated to horseradish peroxidase (HRP) (Appendix A), were used as secondary antibodies. 

### 2.6. Immunofluorescence Staining of Recombinant MARV-GP and MARV-NP

To confirm the expression of recombinant MARV-GP and MARV-NP, VeroE6 cells grown on coverslips were infected with MVA-GP or MVA-NP. The conditions for immunofluorescence staining were published elsewhere [36,49]. To detect MARV-GP or MARV-NP, primary antibodies targeting the GP or the NP of MARV (Appendix A) were used. Polyclonal goat anti-mouse and goat anti-rabbit secondary antibodies (Appendix A) were used to visualize GP- and NP-specific staining by red and green fluorescence, respectively.

### 2.7. Ethics Statement

Female C57BL/6J mice aged 6–9 weeks were obtained from Charles River Laboratories (Sulzfeld, Germany) and housed in individually ventilated cages (IVCs, Techniplast, Buguggiate, Italy) under specified pathogen-free conditions. Mice were distributed into three experimental groups and vaccinated as described below: MVA-MARV-GP (*n* = 10), MVA-MARV-NP (*n* = 10), non-recombinant MVA (*n* = 10), analyzed for cellular responses (*n* = 6), and antibody responses (*n* = 10) as indicated. Mice had access to food and water ad libitum and were allowed to adapt to the facilities for at least one week before vaccination experiments were started. All animal experiments were handled and conducted in compliance with the European and national regulations for animal experimentation (European Directive 2010/63/EU; Animal Welfare Acts in Germany) and Animal Welfare Act, approved by the Regierung von Oberbayern (Munich, Germany).

### 2.8. Vaccination Experiments in Mice

Groups of C57BL/6J mice were immunized twice over a 21-day interval with vaccine suspensions containing 10^7^ PFU recombinant MVA-MARV-GP (MVA-GP), MVA-MARV-NP (MVA-NP), or non-recombinant MVA (MVA) into the left hind leg via the intramuscular (i.m.) route, which correlates to the standard application route of MVA-based vaccines in humans. An immunization dose of 10^7^ PFU per application has proven safe and highly immunogenic in previous mouse immunization studies [36,64,65,66] and was therefore chosen for the current study. After immunization, mice were weighed daily and monitored for clinical abnormalities, including general conditions (e.g., ruffed fur, weight loss) and behavior (e.g., anxious, hyperactive) that had been predefined in the supervision protocol. Mice were euthanized when a humane endpoint was reached or due to experimental settings. Blood was collected on day 35 post prime immunization (final day), and coagulated blood was centrifuged at 500× *g* for 15 min to separate serum, which was stored at −80 °C until further use.

### 2.9. Generation of Peptides, Design of Peptide Pools, and Peptide Prediction

MARV-NP and MARV-GP protein sequences were obtained, and peptides were identified using an in silico approach. All T cell epitope predictions were specific for the MHC class I alleles H2-Db and H2-Kb. For MARV-GP, only predicted peptides were used to identify potential T-cell epitopes. MARV-GP MHC class I (MHC-I)-specific peptides with lengths of 8–11mer were predicted using the Immune Epitope Database (IEDB) tool T Cell Prediction—Class I, using the NetMHCpan 4.1 and the basic processing predictions prediction methods [67]. The results were then screened for peptides with a maximum percentile rank of 5 and a maximum IC_50_ of 500 nM. The peptides identified after the screening were further analyzed for H2-Db and H2-Kb binding using the RankPep tool [68]. For MARV-NP, both predicted and overlapping peptides were used to identify potential T cell epitopes. MARV-NP MHC-I-specific peptides were predicted as described above for MARV-GP. In addition, we designed a set of 15mer peptides with an 11mer overlap that spanned the entire MARV-NP protein sequence (Appendix A).

The following strategies were used to organize peptides for testing. The top three MARV-GP and MARV-NP predicted peptides were tested alone. The remaining predicted peptides were split into two approximately equal-sized pools for testing (Appendix A). Overlapping peptides were organized into pools using a peptide pool matrix system, as described previously [69,70]. All peptides were synthesized by Thermo Fisher Scientific (Planegg, Germany) as crude material (>50% purity) on a 1 mg scale. Peptides were dissolved to 2 mg/mL in either PBS or DMSO under sterile conditions, aliquoted, and stored at −20 °C.

### 2.10. T Cell Analysis by Enzyme-Linked Immunospot (ELISPOT)

On day 35, post prime immunization, mice were euthanized, and splenocytes were isolated, as published elsewhere [36]. ELISPOT assay (ELISPOT Plus Mouse IFN-γ (ALP), Mabtech AB, Nacka Strand, Sweden) was conducted following the manufacturer´s instructions and protocols as described previously [36]. Stimulation of the splenocytes was performed with either individual peptides (2 µg/mL in RPMI supplemented with 10% FBS) or pools of predicted or overlapping peptides (Thermo Fisher Scientific, Planegg, Germany; Appendix A).

### 2.11. T Cell Analysis by Intracellular Cytokine Staining (ICS)

On day 35, after prime immunization, mice were euthanized, and intracellular cytokine staining was conducted as published recently (Appendix A) [36]. Data were acquired using the NovoCyte Quanteon flow cytometer (Agilent Technologies, Waldbronn, Germany) and analyzed using FlowJo version 10.10 (FlowJo LLC, Ashland, OR, USA).

### 2.12. Antigen-Specific IgG Enzyme-Linked Immunosorbent Assay (ELISA)

MARV-GP- and MARV-NP-specific binding antibodies (IgG) were measured by ELISA. Recombinant MARV-GP (The Native Antigen Company, Kidlington, UK) or MARV-NP (Nordic Biosite, Täby, Sweden) were plated at 50 ng/well into 96-well flat bottom ELISA plates (Nunc MaxiSorp Plates; Thermo Fisher Scientific, Planegg, Germany) and incubated overnight at 4 °C. The conditions for ELISA are published elsewhere [36]. As the geometric mean of the optical density (OD) 450 nm values for each individual MVA control at a dilution of 1:100 was less than 0.1, the cut-off for positivity was calculated so that the cut-off OD value was a minimum of 0.2. Therefore, we determined the cut-off value by using mean OD 450 nm value of the MVA control group (dilution 1:100) plus four standard deviations (mean + 4 SD). 

### 2.13. Statistical Analysis

Statistical analyses were performed using Prism 5 (GraphPad Software Inc., Boston, MA, USA). The statistical test was chosen based on the distribution of the data and whether the data set contained outliers. The Mann–Whitney U test was used for the statistical analysis of ELISPOT and ICS data. ELISA optical density (OD) values were calculated as geometric means and log10 transformed before being analyzed by an unpaired two-tailed *t*-test.

## 3. Results

### 3.1. Generation and In Vitro Characterization of Single Recombinant MVA Backbone Viruses Expressing mCherry or GFP

Recombinant MVA backbone viruses sfMVA-mCherry (MVA-mCherry) and sfMVA-GFP (MVA-GFP) were prepared under serum-free conditions, as described in previous studies [36,49,50], allowing us to generate recombinant MVA candidate vaccines suitable for clinical use. The cDNA containing the coding sequences for mCherry and GFP were introduced into the plasmids pLW73 [63] or pG06 to obtain the vector plasmids pLW73-mCherry and pG06-GFP, respectively. The expression of mCherry was placed under transcriptional control of the synthetic VACV-specific early/late promoter PmH5 [61]. In contrast, GFP encoding sequences were placed under transcriptional control of the VACV-specific early/late promoter p11. The well-characterized ancestor virus MVA-F6 [71] served as a backbone virus to introduce the mCherry or GFP encoding sequences by homologous recombination between the open reading frames (ORF) of the two viral genes, *MVA069R* and *MVA070L* (Figure 1a), or deletion site III (Figure 1b), respectively. Clonal recombinant MVA viruses expressing mCherry (MVA-mCherry) or GFP (MVA-GFP) were obtained by serial rounds of plaque purification using GFP or mCherry as a selection marker. MVA-mCherry- or MVA-GFP-infected cell cultures were serial 10-fold diluted and used to re-infect CEF cells growing in 24-well plates. After several rounds of plaque purification, single plaques were collected (Figure 1c), and recombinant MVA-mCherry and MVA-GFP were amplified to high viral titers. Following standardized quality control procedures, we confirmed genetic integrity (Figure 1d,f, Appendix A), replication deficiency in three human cell lines (HaCat, HeLa, A549), and replication efficacy in the producer CEF cells (Figure 1e,g). In addition, we performed Western blot analysis to confirm unimpaired protein expression in A549, CEF, HeLa, and DF-1 cells (Appendix A). 

In summary, we generated and characterized two MVA backbone viruses expressing either mCherry (MVA-mCherry) or GFP (MVA-GFP). This allowed for recombinant MVA candidate vaccines to be developed using red-to-green gene swapping or green-to-red gene swapping, respectively, as a screening method.

### 3.2. Design and Generation of a Recombinant MVA-MARV-GP Candidate Vaccine

To investigate the feasibility of using our MVA backbone virus, MVA-GFP, to accelerate the development of MVA-based vaccines, we generated a recombinant MVA expressing the GP of MARV. The cDNA coding for MARV-GP was placed under transcriptional control of the synthetic VACV-specific early/late promoter PmH5 [61] in the MVA vector plasmid pIIIH5red-MARV-GP and was introduced into deletion site III in the MVA-GFP genome by homologous recombination (Figure 2a), replacing the GFP gene sequence with the MARV-GP-mCherry expression cassette (switch from green-to-red fluorescence). The clonal recombinant MVA viruses were isolated in less than ten rounds of plaque purification using the co-expression of the red fluorescent marker protein mCherry for the facilitated distinction between non-recombinant MVA-GFP (green foci) and recombinant MVA-MARV-GP (red foci) (Appendix A). PCR analysis of the isolated viral DNA from the final recombinant virus demonstrated site-specific insertion of the MARV-GP gene sequence into the MVA genome, proper removal of the marker protein mCherry (Figure 2b, Appendix A) during virus amplification, and genetic stability of the virus (Appendix A). The suitability of the recombinant MVA-MARV-GP for clinical use was confirmed by demonstrating its replicative deficiency in human HaCat cells. In addition, the suitability of recombinant MVA-MARV-GP for manufacturing at an industrial scale was confirmed by demonstrating its replicative capacity in avian CEF producer cells (Figure 2c). To examine the expression of the recombinant GP in more detail, we stained MVA-MARV-GP-infected Vero cells with a mouse monoclonal GP-specific antibody and analyzed it using fluorescence microscopy (Figure 2d). We observed specific GP staining in permeabilized Vero cells, matching the expected intracellular localization of the GP (Figure 2d). In addition, we prepared cell lysates from MVA-MARV-GP-infected Vero cells for immunoblot analysis (Figure 2e,f). By using a mouse monoclonal antibody directed against the GP2 subunit of the recombinant protein, we detected three prominent protein bands, which migrated with molecular masses of ~130 kDa, ~90–95 kDa, and 50–55 kDa (Figure 2e), corresponding to the full-length GP1/2 (glycosylated), pre-GP (glycosylated) [15,72,73], and the GP2 (glycosylated) cleavage product, respectively. The detected protein migrated at molecular masses higher than the ~72 kDa predicted for full-length MARV-GP, indicating that the protein might be glycosylated [15,74]. We treated the cell lysates with peptide-N-glycosidase F (PNGase F), which removes N-linked oligosaccharide chains from different glycoproteins, resulting in reduced molecular masses of the recombinant GP from ~95 kDa to ~72 kDa and from ~55 kDa to ~40 kDa, corresponding to the expected sizes of unmodified MARV-GP and the GP2 cleavage product, respectively (Figure 2e). The expression of full-length MARV-GP and the GP2 cleavage product was already detectable eight hours after infection, indicating proper early transcription driven by the synthetic MVA-specific promoter PmH5 [61] (Figure 2f).

### 3.3. Design and Generation of Recombinant MVA-MARV-NP Candidate Vaccine

To investigate the feasibility of the newly generated MVA-mCherry backbone virus for faster development of MVA-based vaccines, we generated a recombinant MVA vaccine expressing the nucleoprotein (NP) of MARV. The cDNA coding for MARV-NP was placed under transcriptional control of the synthetic VACV-specific early/late promoter PmH5 [61] in the MVA vector plasmid pLW73-MARV-NP and introduced between the open reading frames (ORF) of the two viral genes, *MVA069R* and *MVA070L* (Figure 3a), in the MVA-mCherry genome by homologues recombination, replacing the mCherry gene sequence with the MARV-NP-GFP expression cassette (red-to-green gene swapping). Clonal recombinant MVA viruses were isolated in less than 10 rounds of plaque purification using the co-expression of GFP for facilitated distinction between non-recombinant MVA-mCherry (red foci) and recombinant MVA-MARV-NP (green foci) (Appendix A). PCR analysis of the viral DNA isolated from the final recombinant virus demonstrated the site-specific insertion of the MARV-NP gene sequence into the MVA genome, proper removal of the marker protein GFP (Figure 3b, Appendix A) during virus amplification, and genetic stability of the virus (Appendix A). The suitability of recombinant MVA-MARV-NP for clinical use was confirmed by demonstrating its replicative deficiency in human HaCat cells. Additionally, the suitability of recombinant MVA-MARV-NP for manufacturing at an industrial scale was confirmed by demonstrating its replicative capacity in CEF producer cells (Figure 3c). To examine the expression pattern of recombinant MARV-NP more in detail, we stained MVA-MARV-NP infected Vero cells with a polyclonal antibody directed against the NP and analyzed it by fluorescence microscopy (Figure 3d). We observed specific NP staining in permeabilized Vero cells, consistent with the expected intracellular localization of the NP in cytoplasmic inclusion bodies of variable size. In addition, we prepared cell lysates from MVA-MARV-NP-infected Vero cells for subsequent immunoblot analysis (Figure 3e). Using a monoclonal antibody directed against NP, we detected one prominent band that migrated with a molecular mass of ~100 kDa. The expression of recombinant MARV-NP was already detectable 8 hpi, demonstrating the proper early transcription driven by the synthetic MVA-specific promoter PmH5 [61].

### 3.4. Evaluation of MARV-Specific Immune Responses in C57BL/6J Mice

To perform an initial screening of our two recombinant MVA-MARV vaccine candidates, we vaccinated C57BL/6J mice with 10^7^ PFU using intramuscular (i.m.) administration and a prime-boost immunization schedule over a 3-week interval (Appendix A) and screened for MARV-specific antibody and T cell responses. Mice were monitored and weighed daily, and no clinical abnormalities were observed, demonstrating the tolerability of the two vaccines (Appendix A). 

To analyze the immunogenicity of our MVA-GP and MVA-NP candidate vaccines, we assessed humoral immune responses by IgG ELISA. Serum samples from MVA-GP- and MVA-NP-vaccinated mice, collected at day 35 post prime immunization (14 days after the booster immunization), were tested for GP- and NP-specific binding antibodies. Two immunizations with MVA-GP or MVA-NP resulted in substantial titers of IgG-binding antibodies specific for MARV-GP (Figure 4a) or MARV-NP (Figure 4b), which were significantly higher compared to control-vaccinated mice which received empty MVA-vector. The log_10_ mean binding antibodies were 3.86 ± 0.21 (coefficient of variation (CV) = 16.9%) and 3.19 ± 0.23 (CV = 22.6%) for MVA-GP and MVA-NP vaccinated mice, respectively. 

At the experimental endpoint on day 35, we isolated splenocytes and monitored for GP- and NP-epitope-specific CD8+ T cells using the IFN-γ ELISPOT assay and/or ICS-FACS analysis (Figure 5 and Figure 6). Since only limited information on the antigen specificities of MARV-specific T cells was available, we screened the Immune Epitope Database (IEDB) to identify putative GP- and NP-specific peptide epitopes specific for the C57BL/6J mouse alleles H2-Db and H2-Kb (Appendix A). In addition, we also generated a set of 15mer peptides, overlapping by 11mer, that spanned the entire NP sequence (Appendix A). 

After prime-boost vaccination with MVA-GP, we found that the peptides of GP pool 1 and the individual peptide GP_184–191_ (FSLINRHAI), an H2-Db-restricted peptide, stimulated IFN-γ spot-forming counts (SFC) above the MVA control background (Figure 5a,d). The mean counts were 796 ± 210 SFC/10^6^ cells for GP pool 1 and 134 ± 33 SFC/10^6^ cells for GP_184-191_. The IFN-γ ICS assay confirmed that the peptides mentioned above stimulated GP-specific CD8+ T cells, whereas the other individual peptide and GP pool 2 did not (Figure 5b,e). The frequency of IFN-γ-producing antigen-specific CD8+ T cells were 0.11 ± 0.03% for GP_184-191_ and 0.74 ± 0.16% for GP pool 1. When we analyzed for the co-production of TNF-α, we observed that the majority of GP pool 1 and GP_184-191_ stimulated CD8+ T cells produced both IFN-γ and TNF-α (88 ± 3.2% and 50 ± 9.4% for GP pool 1 and GP_184-191_ respectively) (Figure 5c), suggesting that the vaccine stimulated GP-specific polyfunctional CD8+ T cells.

Our mouse immunization study also demonstrated that MVA-NP vaccination activated NP-specific CD8+ T cells. The screening of 15mer overlapping peptides revealed 7 pools that had significantly higher IFN-γ SFCs s than the MVA control group (Figure 6a). The top three of those peptide pools were V11, H3, and V5, which had mean counts of 235 ± 59 SFC/10^6^ cells, 231 ± 33 SFC/10^6^ cells, and 87 ± 42 SFC/10^6^ for the MVA-NP group, respectively. In addition, we tested two pools containing peptides predicted by IFN-γ ELISPOT assay and found that one pool, NP pool A2, stimulated antigen-specific T cells with a mean count of 80 ± 15 SFC/10^6^ cells (Figure 6b). In the same experiment, the top three predicted peptides were also tested individually by IFN-γ ELISPOT assay and ICS-FACS analysis. The H2-Db-restricted peptides NP_32-42_ (VSICNQIIDAI) and NP_463-473_ (FALLNEDEDTL) had significantly higher counts than the MVA control group by IFN-γ ELISPOT assay, although the trend was not observed with ICS-FACS analysis. This discrepancy was likely due to the reduced sensitivity of the ICS assay relative to the IFN-γ ELISPOT assay. Moreover, due to the low frequency of IFN-γ producing cells in the ICS assay, the cytokine profile of the MARV-NP-specific CD8 T cell population could not be analyzed. The mean ELISPOT counts of the MVA-NP group were 76 ± 14 SFC/10^6^ for the NP_32-42_ group and 74 ± 14 SFC/10^6^ cells for NP_463-473_. Interestingly, neither of these peptide sequences were found within the 15mer overlapping peptides of the top three peptide pools (V11, H3, and V5). Finally, when we compared the vector-specific T cell immunity induced by MVA-GP and MVA-NP to non-recombinant MVA (MVA), we found no differences in the magnitude of the CD8+ T cell response (Appendix A).

Taken together, our results indicate that the recombinant MVA viruses MVA-GP and MVA-NP generated strong humoral and T cell-mediated immune responses after prime-boost immunization. Importantly, they induced robust antigen-specific T cell immunity, with the MVA-GP candidate vaccine stimulating polyfunctional antigen-specific CD8+ T cells. In the next step, we aim to investigate the protective capacity of our recombinant MVA viruses.

## 4. Discussion

Several smaller outbreaks of MARV in humans have occurred in different African regions over the last few years. MARV is highly contagious, with mortality rates up to 90%, which underlines the need for rapid public health measures in the case of an outbreak. The availability of a MARV vaccine for humans should significantly reduce the risk of transmission and severe disease outcomes. Previously, a ChAd3-based MARV vaccine induced robust immunogenicity and protective efficacy in macaques [75], and it is now in clinical evaluation studies in humans [76]. Due to the MARV outbreak in Rwanda, the local health authorities started administering vaccine doses to combat the outbreak [77]. This immunization campaign is part of a phase 2 open-label trial from the Sabin Vaccine Institute (Washington, DC, USA). Several other MARV candidate vaccines are being evaluated in preclinical and clinical settings. However, none have received approval for human use (for review: [78]) to date. Small animal models are preferred over non-human primates due to their low cost and availability. However, the protective capacity observed in laboratory mice or guinea pigs is not always reproducible when using NHPs. In addition, the limited commercial market and the resulting lack of funding and detailed information about the correlation of protection hamper the development of effective MARV vaccines [78]. We intended to use replication-deficient MVA as a viral vector to express MARV antigens. 

The use of MVA as a viral vector for the generation of MARV vaccines is further supported by the promising results of other MVA-based vaccines against emerging viruses [34,35,36,79]. In these previous studies, recombinant MVAs against MERS-CoV and SARS-CoV-2 have been successfully tested in phase I clinical studies in humans, demonstrating safety and immunogenicity [35,80]. In addition, we also successfully generated recombinant MVAs expressing either EBOV-GP or EBOV-NP, which induced robust immunogenicity and efficacy in a mouse model for lethal EBOV disease [34]. Based on our experience with the MVA-EBOV vaccines, we decided to use MARV-GP and MARV-NP as immunogens to be expressed by MVA.

An important requirement to efficiently combat new or re-emerging viruses is that the respective vaccine is rapidly available in the case of an outbreak. MVA is safe and activates a strong immune response, including in elderly and immunocompromised patients. Repeated MARV outbreak situations show that a potential vaccine candidate needs to be easy to store for emergency use and should have a long shelf-life. This has been identified as a disadvantage of mRNA-based vaccines that must be stored at low temperatures, as they are unstable at room temperature. In contrast, MVA-based vaccines are very stable, which allows for less problematic distribution and deployment in warmer countries such as those in Africa.

Plaque passages to clonally isolate recombinant MVAs are always hampered by non-recombinant MVA parental virus. To further speed up the plaque purification process, we adapted our color screening methodology to specifically distinguish the MVA backbone virus from the clonal recombinant MVA virus so that the clonal isolation process is more effective in eliminating non-recombinant MVA parental virus. The use of fluorescent MVA backbone viruses that meet requirements for the production of recombinant MVA candidate vaccines under pre-GMP conditions has two major advantages: on the one hand, we were able to halve the time that is needed to isolate a positive clonal isolate compared to the usage of a non-fluorescent MVA backbone virus. On the other hand, the established workflow used to generate and characterize MVA candidate vaccines based on the MVA backbone viruses allow direct shipping to manufacturing companies, thus saving several weeks of laboratory work to amplify the viruses under pre-GMP conditions.

For this, we generated parental MVAs expressing a different fluorescent reporter protein than that inserted within the MVA transfer plasmid (sfMVA-GFP, sfMVA-mCherry). Since the fluorescent reporter protein is expressed at the same site targeted by the MARV-GP and MARV-NP gene sequences expressed within the transfer plasmids, we confirmed successful homologous recombination by a switch from red to green fluorescence, or vice versa. In addition, non-recombinant MVA viruses which are always present during plaque passages to isolate the clonal recombinant MVA are indicated by the respective fluorescence expression. In this study, we successfully generated appropriate parental serum-free MVA-viruses using the clonal isolate that was established for clinical use [49]. Again, we used our well-established standard methodology compatible with clinical evaluation in humans to insert the gene encoding sequence for the GFP within deletion III of the serum-free MVA (sfMVA-GFP). The mCherry gene-encoding sequence was successfully inserted within the intergenic sites *MVA069R* and *MVA070L* of serum-free MVA (sfMVA-mCherry).

In this study, we successfully used sfMVA-GFP for the insertion of MARV-GP within deletion site III and sfMVA-mCherry for the insertion of the MARV-NP within the intergenic sites *MVA069R* and *MVA070L* of MVA. Indeed, the use of these parental viruses, including additional fluorescence markers, significantly reduced the number of plaque passages needed for isolating recombinant MVA-MARV-GP and MVA-MARV-NP from about 20 to less than 10 passages, which significantly accelerated the generation of MVA-based vaccine candidates. Our experiments confirmed the genetic stability of all generated vector viruses, and comparative growth analyses confirmed that recombinant MVA–MARV-GP and MVA-MARV-NP replicated in CEF cells as efficiently as the non-recombinant wild-type MVA.

Moreover, we confirmed the stable expression of the MARV-GP and MARV-NP antigens by Western blot (WB) analysis and immunofluorescence (IFA). Of note for MVA-MARV-GP, we also confirmed the authentic procession of the GP. This is demonstrated in the WB analysis by the cleavage from pre-GP into prospective GP1 and GP2 at different time points. Authentic processing is further supported by the glycosylation of MARV-GP, as demonstrated by the use of PNGase treatment in the WB. This aligns with data from previous studies evaluating the assembly and formation of MARV particles [81,82]. The structural protein NP is the driving force for the formation of inclusion bodies where filoviral transcription and replication take place [82]. In our study, we confirmed the formation of NP-mediated inclusion bodies by IFA as well as correct and stable NP expression by MVA, using WB analysis over a time kinetic of 48 h. 

Moreover, in vivo experiments in C57BL/6J mice demonstrated the safety and immunogenicity of the MVA-MARV-GP and MVA-MARV-NP candidate vaccines. We detected the substantial activation of MARV-GP-specific humoral immune responses for the GP-antigen after two vaccinations. Since MARV-GP is responsible for viral attachment and entry, GP-specific antibodies have been considered important to preventing a MARV infection. Since virus-neutralizing antibodies are still considered a correlate of vaccine-induced protection [83], they will be of great interest for further characterizing the neutralizing capacities of these MARV-GP-specific antibodies. The protective efficacy of MARV-neutralizing antibodies has been confirmed using human sera from survivors with high titers of neutralizing antibodies for passive serum transfer in mice with lethal MARV-challenge infection [84]. We also detected the activation of GP-specific T cells after prime-boost MVA-MARV-GP vaccination. This further supported the activation of a balanced immunity, including antibodies and T cells. Interestingly, the quality of the T cell response is mainly directed against epitopes located in the GP1-subunit, since stimulation with pools of predicted peptides covering the GP1-subunit induced significantly higher levels of T cells, as measured by ELISPOT compared to the GP2-subunit. In line with these data, stimulation with the GP_184-191_, which is located in the GP1-subunit, induced significantly higher levels of CD8+ T cells in ELISPOT and also in FACS by ICS compared to other GP-specific peptides covering the GP2-subunit (GP_490-498_, GP_583-591_). The simultaneous production of IFN-γ and TNF-α indicates that the quality of these MARV-GP1-specific CD8+ T cells is directed towards a Th1-specific immune response. So far, little data are available on the activation of MARV-specific T-cell responses in humans. A recent study confirmed the strong activation of CD4+ T cell responses but lower CD8+ T cell responses in MARV survivors, which correlates with a robust activation of MARV-specific antibodies [85]. A robust activation of Th1-specific cellular immune responses by vaccination is also advantageous concerning the beneficial activation of B cells for the production of IgM and IgG. This is further supported by the strong activation of GP-binding IgG-antibodies after prime-boost vaccination, which is considered advantageous with regard to efficient preventive measures against filovirus-induced disease [86].

Traditionally, viral surface proteins were considered to be the classical vaccine antigen since their prominent exposure on the cell surface of viruses and their role in viral morphogenesis and the life cycle have been shown to be the major targets of neutralizing antibodies. However, over the last years, additional structural proteins have been successfully used for the generation of improved and innovative vaccination strategies. For filoviruses, this includes the VP40 matrix protein which, in combination with the expression of the GP, results in the formation of virus like particles (VLPs) that are known for their extraordinarily strong immunity [87]. In addition, the nucleoprotein has been increasingly included as a vaccine antigen [88]. The filovirus NP is highly conserved among many different filoviruses, including different species of Ebola virus [89] and MARV. Thus, the activation of NP-specific antibodies and T cells is considered a promising approach for the induction of a broadly protective filovirus immune responses. The broadly protective efficacy of NP-specific immune responses has been already confirmed in other virus families including orthomyxoviruses, e.g., influenza virus, and *Paramyxoviridae,* e.g., Sendai virus [90]. For influenza virus, a range of different vaccination strategies have targeted the NP for the development of a universal Flu vaccine [91]. Recent studies further support the cross-reactive capacity of the filovirus NP, as sera from bats cross-reacted with NPs from different Ebola virus and MARV-strains [92]. The NP’s role as a cross-reactive filovirus antigen has been further confirmed in a recent study that evaluated sera from Marburg and Ebola survivors 1–14 years after the infection [93]. The efficacy of NP-specific immune responses has already been confirmed in the context of other filovirus-specific vaccines [34]. In this study, we measured the activation of robust titers of NP-specific immune responses after two vaccinations with MVA-MARV-NP. Again, it will be of interest to evaluate the protective capacity of MARV-NP-specific immune responses in the context of lethal MARV challenge infection. In previous studies, we also generated an MVA-based vaccine against Ebola virus (EBOV) targeting the EBOV-NP from the Mayinga strain (MVA-EBOV-NP). MVA-EBOV-NP robustly protected against lethal EBOV challenge infection in a lethal EBOV mouse model. The protective efficacy of the MVA-EBOV-NP vaccine could be associated with the robust activation of NP-specific antibodies and NP-specific T cells also. Of note, MVA-EBOV-NP-mediated protection could be correlated to the presence of NP-specific CD8+ T cells, since the depletion of the CD8+ T cells completely abrogated protective efficacy after vaccination [34]. In addition, MVA-MARV-NP vaccination also resulted in the activation of high levels of MARV-NP-specific T cells. Here, we used two matrix-based pools of overlapping 15mer peptides that span the entire NP protein to identify highly reactive NP T cell epitopes. In doing so, different potential epitopes at different sites within the NP sequence were shown to induce the activation of IFN-γ-secreting T cells. When we further mapped the specificities of these peptides, we narrowed down a highly specific reaction in the N-terminal region, as shown by the significant activation of IFN-γ-secreting T cells when using the predicted NP_32-42_ and NP_463-473_ peptides. Following on from this, it will be of great interest to evaluate the protective efficacy of a monovalent MVA-MARV-NP vaccine that is mainly associated with the activation of NP-specific T cells and non-neutralizing antibodies. In addition, it is important to further evaluate the protective efficacy of the MVA-MARV-GP vaccine and to compare it with a multivalent MVA-based vaccine expressing the GP and the NP in one vaccine. Since the results from the current study also support the application of the fluorescence reporter proteins for very convenient and rapid screening, modifications and adaptations of such MVA-based vaccines are also possible regarding new emerging filovirus species. This is an advantage in the case of an outbreak, during which a vaccine must be available in a very short time. Moreover, the results from the immunogenicity testing of the MVA-MARV-GP, and MVA-MARV-NP candidate vaccines in C57BL/6J mice for safety and immunogenicity further support more detailed characterization of single vaccination strategies and short-term vaccination approaches. Moreover, the evaluation of the protective efficacy and the correlation of vaccine-induced protection to selected components of the adaptive immune system will be of interest. To overcome the limitations of mouse studies, larger studies in non-human primates will contribute to better assess the translational efficacy of MVA-MARV-GP and MVA-MARV-NP vaccine candidates. These results will further support the advanced clinical development of MVA-based vaccines for applications in humans. Future clinical studies in humans will address limitations from animal studies regarding variability in responses and challenges in translating to human models.

## 5. Conclusions

In conclusion, our data demonstrated the usability of sf-MVA-GFP and sf MVA-mCherry to more rapidly generate recombinant MVAs to be used as vaccines against new emerging viruses. As a proof of concept, we used sfMVA-GFP and sfMVA-mCherry to generate MVA-based vaccines against MARV in half the time. The newly generated MVA-MARV-GP and MVA-MARV-NP proved safe and immunogenic in mice. Based on these results, more preclinical studies will be performed to evaluate the protective efficacy against lethal MARV challenge infection. These data will further support clinical studies in humans to assess the safety and immunogenicity of the MVA-MARV vaccine candidates in the future.

## Figures and Tables

**Figure 1 vaccines-12-01316-f001:**
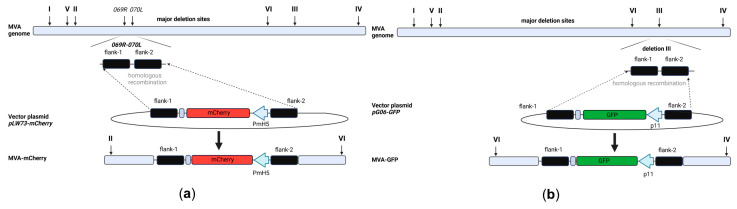
Design, construction, and virological in vitro characterization of sfMVA-mCherry (MVA-mCherry) and sfMVA-GFP (MVA-GFP). (**a**,**b**) Schematic representation of the MVA genome. The six major deletion sites are indicated with I-VI. The encoding sequences of mCherry (**a**) and GFP (**b**) were introduced into non-recombinant MVA by homologous recombination. mCherry was inserted into the genomic region between the two MVA genes, *MVA069R* and *MVA070L*, within the MVA genome. GFP was inserted into deletion III within the MVA genome. PCR analysis was conducted by using a specific oligonucleotide primer targeting the respective insertion site. (**c**) Fluorescent microscopy of recombinant MVA-mCherry and MVA-GFP after serial round of plaque purification. Scale bar: 50 µm. (**d**) Genetic integrity of MVA-mCherry. PCR analysis of viral DNA with oligonucleotide primers specific for the target site confirmed the insertion of full-length mCherry gene sequence. (**e**) Viral growth profile of MVA-mCherry and non-recombinant MVA. (**f**) Genetic integrity of MVA-GFP. PCR analysis of viral DNA with oligonucleotide primers specific for the target site confirmed the insertion of a full-length GFP gene sequence. (**g**) Viral growth profile of MVA-GFP and non-recombinant MVA (MVA). hpi: hours post-infection.

**Figure 2 vaccines-12-01316-f002:**
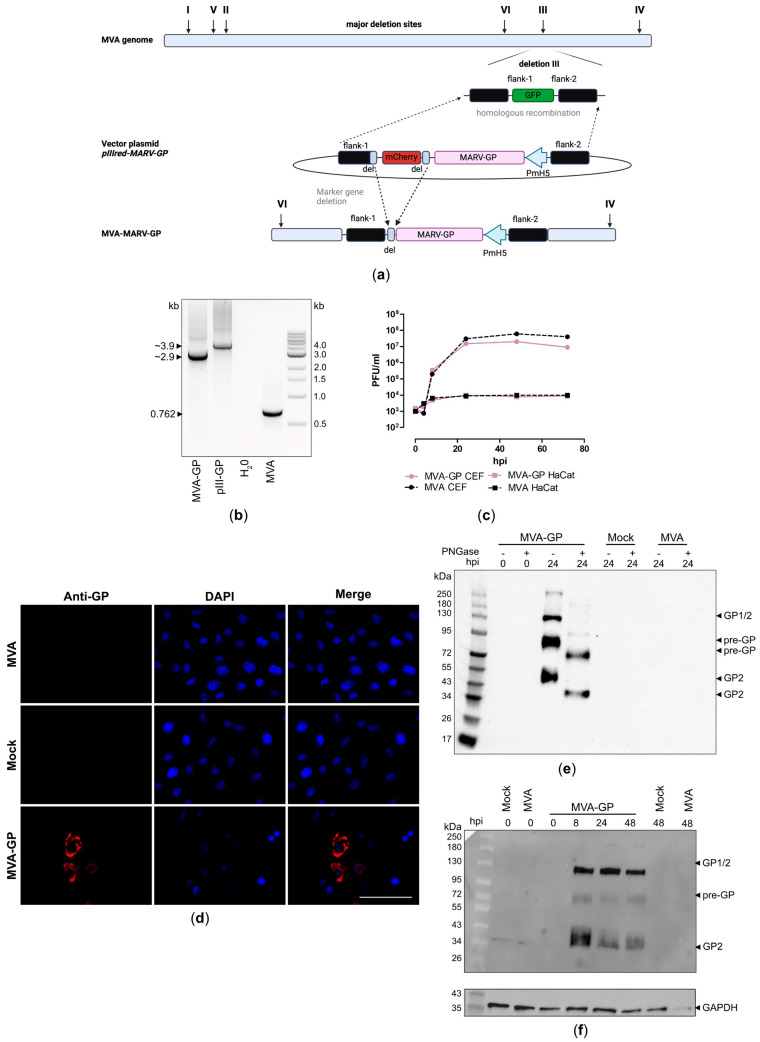
Design, construction, and virological in vitro characterization of MVA-MARV-GP (MVA-GP) vaccine candidate. (**a**) Schematic representation of sfMVA-GFP (MVA-GFP) genome. The codon-optimized full-length MARV-GP gene sequence was inserted into MVA-GFP by homologous recombination. Removal of marker gene mCherry occurred by intragenomic recombination during plaque purification. (**b**) The genetic integrity of MVA-GP was determined by PCR analysis using an oligonucleotide primer specific for the targeted insertion site (deletion site III). (**c**) Viral growth profile of MVA-GP and non-recombinant MVA (MVA). (**d**–**f**) Expression of full-length MARV-GP in MVA-GP infected VeroE6 cells, analyzed by (**d**) immunofluorescence staining and (**e**,**f**) Western blot. (**d**) Permeabilized VeroE6 cells were probed with a primary antibody targeting MARV-GP and a polyclonal goat-anti-mouse secondary antibody for GP-specific staining (red). DAPI served to counterstain cell nuclei (blue). Scale bar: 50 µm. (**e**,**f**) Proteins in cell lysates of MVA-GP-infected VeroE6 cells were separated by SDS-PAGE and subsequently probed with a primary antibody targeting MARV-GP (GP2 subunit). (**f**) The blot was incubated with a primary antibody targeting GAPDH to confirm the loading of equal amounts of proteins (**e**) Deglycosylation of cell lysates was performed with PNGase F before Western blot analysis. Uninfected cells (mock) and cells infected with non-recombinant MVA (MVA) served as controls. hpi: hours post-infection.

**Figure 3 vaccines-12-01316-f003:**
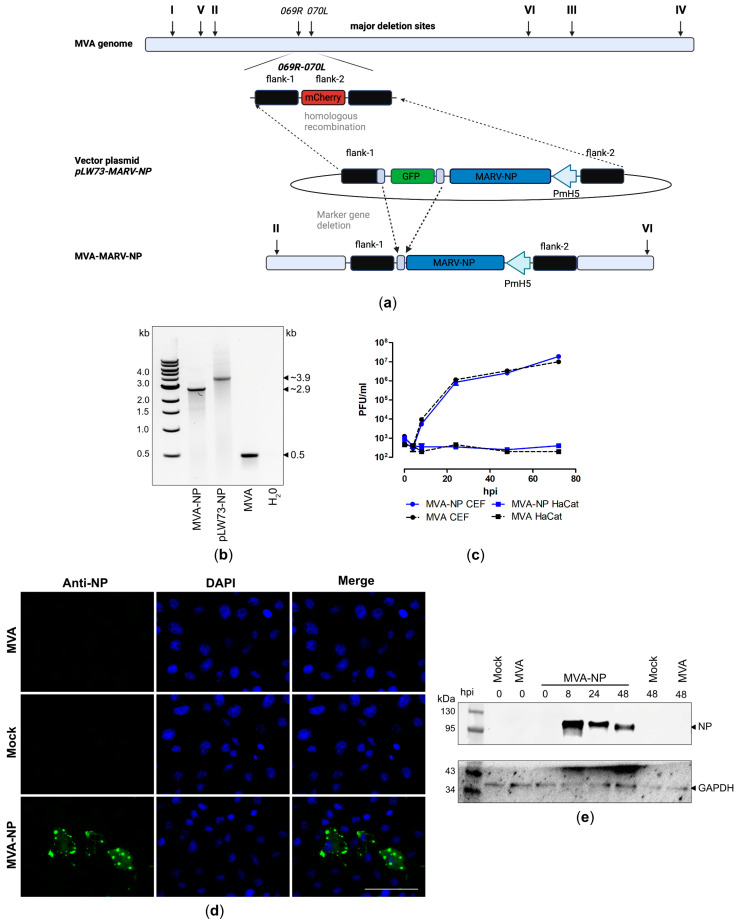
Design, construction, and virological in vitro characterization of MVA-MARV-NP (MVA-NP). (**a**) Schematic representation of the sfMVA-mCherry genome. The codon-optimized full-length MARV-NP gene sequence was inserted into MVA-mCherry by homologous recombination. Removal of the marker gene GFP occurred by intragenomic recombination during plaque purification. (**b**) Genetic integrity of MVA-NP was determined by PCR analysis using an oligonucleotide primer specific for the targeted insertion site (intragenomic region between *MVA069R* and *MVA070L*). (**c**) Viral growth profile of MVA-NP and non-recombinant MVA (MVA). (**d**,**e**) Expression of full-length MARV-NP in MVA-NP infected VeroE6 cells, analyzed by (**d**) immunofluorescence staining and (**e**) Western blot. Permeabilized Vero cells were probed with a primary antibody targeting MARV-NP and a polyclonal goat-anti-rabbit secondary antibody for NP-specific staining (green). DAPI served for counterstaining cell nuclei (blue). Scale bar: 50 µm. (**e**) Proteins in cell lysates of MVA-NP infected Vero cells were separated by SDS-PAGE and subsequently probed with a primary antibody targeting MARV-NP. Uninfected cells (mock) and cells infected with non-recombinant MVA (MVA) served as controls. hpi: hours post infection.

**Figure 4 vaccines-12-01316-f004:**
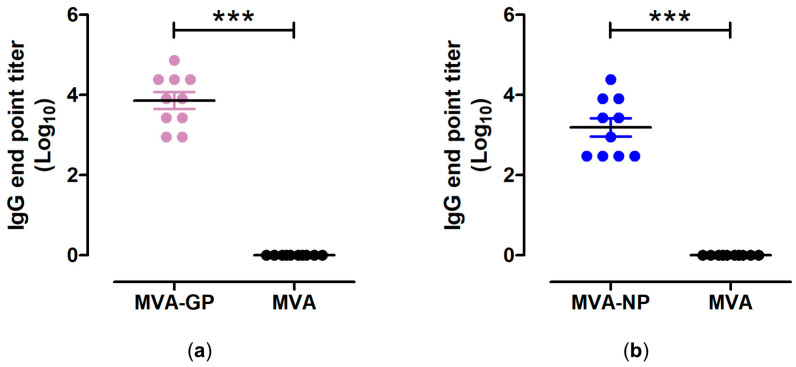
Antigen-specific humoral immunity induced by MVA-MARV-GP (MVA-GP) and MVA-MARV-NP (MVA-NP). Groups of C57BL/6J mice (n = 10) were i.m. immunized twice with 10^7^ PFU MVA- GP or MVA- NP. Serum samples were collected at day 35 after prime immunization and analyzed for MARV-GP- (**a**) or MARV-NP- (**b**) specific IgG binding titers by ELISA. Geometric means were calculated and data were log transformed and analyzed by unpaired two-tailed *t* test. Bars shows the mean + standard error of the mean (SEM). *** *p* < 0.001.

**Figure 5 vaccines-12-01316-f005:**
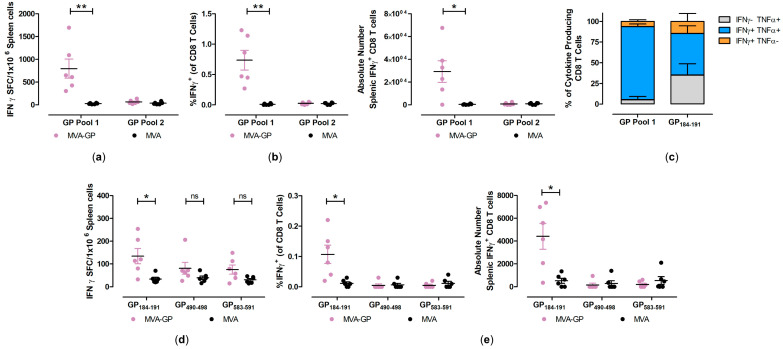
Activation of GP-specific cellular immune responses after prime-boost immunization with MVA-MARV-GP (MVA-GP). Groups of C57BL/6J mice (n = 6) were i.m. immunized twice with 10^7^ PFU MVA-GP. Mice immunized with non-recombinant MVA served as controls. Splenocytes were collected and prepared on day 35 after prime immunization. (**a**–**c**) Splenocytes were stimulated with pools containing H2-Db- and H2-Kb-restricted peptides and analyzed using IFN-γ ELISPOT assays and IFN-γ/TNF-α ICS plus FACS analysis. (**d**,**e**) Splenocytes were stimulated with the individual peptides GP_184-191_, GP_490-498_, GP_583-591_, and were tested using IFN-γ ELISPOT assays and IFN-γ/TNF-α ICS plus FACS analysis. (**b**,**e**) IFN-γ producing CD8+ T cells measured by FACS analysis. Graphs show the frequency and absolute number of IFN-γ+ CD8+ T cells. (**c**) Cytokine profile of MARV-GP-specific CD8 T cells. Graphs show the mean frequency of IFN-γ-TNF-α+, IFN-γ+TNF-α+, and IFN-γ+TNF-α- cells within the cytokine positive CD8 T cell compartment and the error bars represent the standard error of the mean (SEM). Bars in scatter plots represent the mean + SEM. For each peptide or peptide pools tested by IFN-γ ELISPOT and ICS, the MVA-GP and MVA groups were analyzed by two-tailed Mann–Whitney U test. ns not significant, * *p* < 0.05, ** *p* < 0.01.

**Figure 6 vaccines-12-01316-f006:**
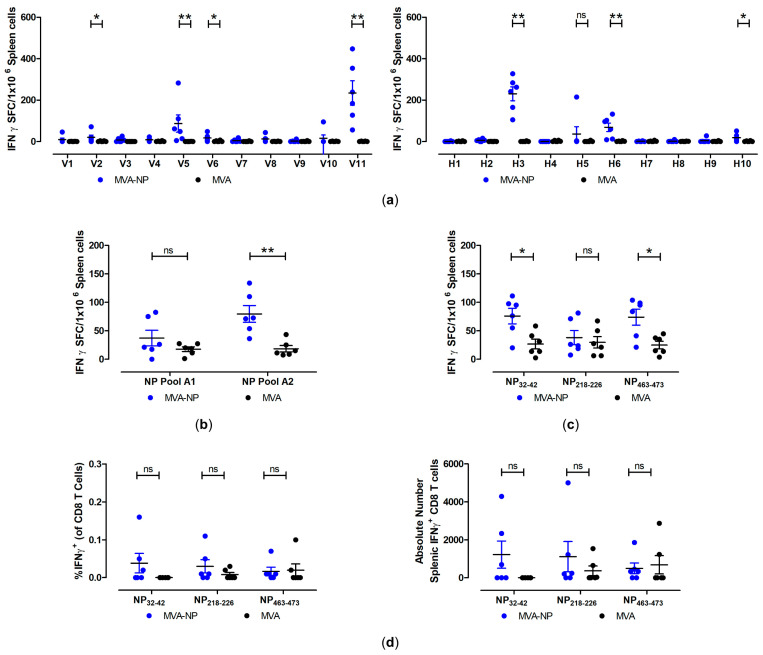
Activation of NP-specific cellular immune response after prime-boost immunization with MVA-MARV-NP (MVA-NP). Groups of C57BL/6 mice (n = 6) were i.m. immunized twice with 10^7^ PFU MVA-MARV-NP. Mice immunized with non-recombinant MVA served as controls. Splenocytes were collected and prepared on day 35 after prime immunization. (**a**,**b**) Splenocytes were stimulated with pools containing H2-Db- and H2-Kb-restricted peptides and tested using IFN-γ ELISPOT assay. (**c**,**d**) Splenocytes were stimulated with the individual peptides NP_32-42_, NP_218-228_, NP_463-473_, and were tested using IFN-γ ELISPOT assays and IFN-γ/TNF-α ICS plus FACS analysis. Bars in scatter plots represent the mean + the standard error of the mean (SEM). For each peptide or peptide pools tested by IFN-γ ELISPOT and ICS, the MVA-NP and MVA groups were analyzed by two-tailed Mann–Whitney U test. ns not significant, * *p* < 0.05, ** *p* < 0.01.

## Data Availability

The original contributions presented in this study are included in the article and Appendix A, and further inquiries can be directed to the corresponding author.

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
