# Peer review of "Rapid Development of Modified Vaccinia Virus Ankara (MVA)-Based Vaccine Candidates Against Marburg Virus Suitable for Clinical Use in Humans"

_vaccines, 2024, doi:10.3390/vaccines12121316_

Round 1
Reviewer 1 Report
Comments and Suggestions for Authors
The results of the study are interesting and may be useful in the development vaccine candidates against Marburg virus. However, I have the following comments to be addressed.
-Line 70. The word Orthomarburgviruses should be written in normal font, not italics. In this case, it is a common name, not a systematic one. The same situation with word Orthomyxoviruses in line 583.
-Lines 70-71. It may also be worth mentioning that the genome is represented by (-) RNA.
-Lines 80-81. The sentence should be rephrased. The information that mRNA vaccines were the first available vaccines is not entirely correct.
-Line 172. The abbreviation “hpi” should be deciphered in the text and in the captions to the figures
-Line 200. Shouldn’t this be “on day 35 post prime immunization” instead of “on day 35”?
-Lines 242-243. Why is the mean + 4SD used as the cut off and not the mean + 2SD or mean + 3SD, or 2SD/3SD of the mean. Please explain the choice of cut off. I would like to see a more detailed description of how the cut off was calculated, as well as how the control and experimental sera were applied to the plates.
-Line 248. Information about the statistical tests used should be checked and adjusted. There is some discrepancy with the captions to the figures.
-Line 280. line 376. “in vitro” should be written in italics
-Lines 286-287. Information about the size of the bar should be transferred to the description of part (c) of Figure 1.
-Figure 1d, 1f. Authors should clarify why the MVA size is different in these figures (0.5 kb and 0.762 kb).
-Figure 2b. Shouldn't there be "pIII-GP" instead of "pIII-GFP"?
-Figure 3e. It seems to me that it is worth indicating in the caption about the anti-GAPDH treatment and why this was done (despite the fact that the materials and methods mention loading control).
-Figure 5. I did’t find that figure 5F was described in the caption.
-In all figures where the word “mock” appears it should be clarified in captions what is represented as “mock”
-It is not clear why sometimes the authors indicate in the figures that there are no statistically significant differences between the groups, but sometimes they don’t indicate. Please explain.
-All figure captions should indicate what the error bars represent.
-Lines 317-318. According to the text, pre-GP is a non-glycosylated form with a molecular weight of 95 kDa. But the authors then write that after treatment with glycosidase, a decrease in the molecular weight of recombinant GP from 95 kDa to 72 kDa is observed (line 324). But isn't the product with a mass of 95 kDa non-glycosylated? Please clarify this point.
-Line 544-545, line 615. I didn't find any safety data in the results section.
Author Response
Reviewer 1:
- The results of the study are interesting and may be useful in the development vaccine candidates against Marburg virus. However, I have the following comments to be addressed.
A: We greatly appreciate the reviewer´s overall positive response.
- Line 70. The word Orthomarburgviruses should be written in normal font, not italics. In this case, it is a common name, not a systematic one. The same situation with word Orthomyxoviruses in line 583.
A: Thank you for the hint. We wrote the words Orthomarburgviruses and Orthomyxoviruses in normal font as suggested.
- Lines 70-71. It may also be worth mentioning that the genome is represented by (-) RNA.
A: We added the information about the genome polarity.
- Lines 80-81. The sentence should be rephrased. The information that mRNA vaccines were the first available vaccines is not entirely correct.
A: Thank you for the hint. We rephrased the sentence.
- Line 172. The abbreviation “hpi” should be deciphered in the text and in the captions to the figures
A: Thank you for the suggestion. We deciphered the abbreviation “hpi” in the text and the figure legends.
- Line 200. Shouldn’t this be “on day 35 post prime immunization” instead of “on day 35”?
A: Thank you for the hint. We added “post prime immunization” to clarify the collection date.
- Lines 242-243. Why is the mean + 4SD used as the cut off and not the mean + 2SD or mean + 3SD, or 2SD/3SD of the mean. Please explain the choice of cut off. I would like to see a more detailed description of how the cut off was calculated, as well as how the control and experimental sera were applied to the plates.
A: The mean OD value for the MVA control group at 1/100 dilution was 0.041. If we used the mean + 2SD or + 3SD as a cut-off value, the cut-off would have been at 0.15 or 0.1, respectively. We prefer to be more conservative with our cut off values. The cut off value for mean + 4SD was 0.2.
- Line 248. Information about the statistical tests used should be checked and adjusted. There is some discrepancy with the captions to the figures.
A: Thank you for the suggestion. We added detailed information about the statistical tests as suggested.
- Line 280. line 376. “in vitro” should be written in italics
A: Thank you for the hint. We wrote “in vitro” in italics.
- Lines 286-287. Information about the size of the bar should be transferred to the description of part (c) of Figure 1.
A: Thank you for the hint. We transferred the description of the bar size to Figure 1c.
- Figure 1d, 1f. Authors should clarify why the MVA size is different in these figures (0.5 kb and 0.762 kb).
A: Figure 1d shows the construction of MVA-mCherry with the encoding sequence of mCherry being inserted between the two MVA genes MVA-069R and MVA-070L. Figure 1f shows the construction of MVA-GFP with the encoding sequence of GFP being inserted into deletion site III of MVA. Due to the different oligonucleotide primers that are used to amplify the insertion region, the amplified amplicons differ in size. We clarified this in the figure caption.
- Figure 2b. Shouldn't there be "pIII-GP" instead of "pIII-GFP"?
A: We apologize for the mistake. We changed the figure accordingly.
- Figure 3e. It seems to me that it is worth indicating in the caption about the anti-GAPDH treatment and why this was done (despite the fact that the materials and methods mention loading control).
A: Thank you for the suggestion. We added information about the anti-GADPH treatment and rationality in the Figure caption.
- Figure 5. I did’t find that figure 5F was described in the caption.
A: We changed the figure accordingly.
- In all figures where the word “mock” appears it should be clarified in captions what is represented as “mock”
A: Thank you for the useful suggestion. Mock represents in all our experiments untreated/uninfected cells. We clarified this in the figure captions were applicable (lines 406-407, 451-452).
- It is not clear why sometimes the authors indicate in the figures that there are no statistically significant differences between the groups, but sometimes they don’t indicate. Please explain.
A: We only showed no statistically significant differences in instances where the mean ELISPOT count or percentage/absolute number of IFN-γ+ cells of the vaccine group was greater than the mean of the control group and the difference was not statistically significant. We also added 'ns' to the graph in instances where it was clear that the peptide stimulated T cells in both vaccine and control groups (e.g., Supplementary Figure S5, vector-specific immunity). If the means were at around 0 for both groups, then we did not add 'ns' to that data set on the graphs because the peptides did not stimulate any antigen-specific T cells (e.g., figures 5 and 6).
- All figure captions should indicate what the error bars represent.
A: Done
- Lines 317-318. According to the text, pre-GP is a non-glycosylated form with a molecular weight of 95 kDa. But the authors then write that after treatment with glycosidase, a decrease in the molecular weight of recombinant GP from 95 kDa to 72 kDa is observed (line 324). But isn't the product with a mass of 95 kDa non-glycosylated? Please clarify this point.
A: We are sorry for this mistake and appreciate the reviewer's attentiveness. The MARV-GP is heavily glycosylated, as Volchkov and colleagues described (Volchkov, Virology, 1991). As identified by Volchov et al., the glycosylation is mainly in the variable, extremely hydrophilic middle region of GP, which explains the change of molecular weight we observed after treatment with glycosidase. We changed the description in the results section appropriately.
- Line 544-545, line 615. I didn't find any safety data in the results section.
A: We weighed and monitored the animals daily, and we did not observe changes in body weight (Supplementary Figure S4b) or clinical abnormalities (e.g., behavior or general conditions) that had been predefined in our supervision protocol and received approval by the local regulation authorities. Due to these observations, we concluded safety and well tolerability of our vaccines.

Reviewer 2 Report
Comments and Suggestions for Authors
According to the manuscript, I’ll provide feedback according to the outlined criteria: the experimental design is comprehensive, including in vitro and in vivo evaluations that examine vaccine candidates’ genetic stability, protein expression, and immunogenicity in mice. However, to enhance the design: please consider the protective efficacy of the vaccine.
Overall, the manuscript is well-constructed, and the study is compelling in the context of rapid vaccine development for MARV.
Author Response
Reviewer 2:
- According to the manuscript, I’ll provide feedback according to the outlined criteria: the experimental design is comprehensive, including in vitro and in vivo evaluations that examine vaccine candidates’ genetic stability, protein expression, and immunogenicity in mice. However, to enhance the design: please consider the protective efficacy of the vaccine. Overall, the manuscript is well-constructed, and the study is compelling in the context of rapid vaccine development for MARV.
A: We greatly appreciate the reviewer´s overall positive response. We agree with the reviewer´s suggestion that evaluating the protective efficacy of the vaccines would significantly improve the study design. The results from such an efficacy study will be significant for developing a safe and effective MARV-vaccine candidate. However, since MARV is a biosafety 4 agent, the protective efficacy cannot easily be evaluated. The biosafety requirements to work with BSL-4 agents are beyond our animal facility. For this, the protective efficacy of our MVA-MARV candidate vaccines was beyond the scope of this study, and this question will be addressed more in detail in future studies. The main goal of this study was to demonstrate the suitability of our two backbone viruses, MVA-GFP and MVA-mCherry, for rapid deployment of candidate vaccines suitable for clinical use. We used the Marburg virus as an example to prove our concept.

Reviewer 3 Report
Comments and Suggestions for Authors
The manuscript titled, “Rapid development of Modified Vaccinia virus Ankara (MVA)-based vaccine candidates against Marburg virus suitable for clinical use in humans,” details the authors efforts to quicken the development process of MVA vectored vaccines for Marburg virus, and perhaps other viral threats. It is a well written manuscript that accurately documents the research conducted to insert tools in the vector background that will help streamline vector production. The authors use Marburg as their test candidate and show relevant immune responses in mice after vaccination compared to MVA. The only critique I can offer is that there is a general lack of impact regarding the purpose laid out by the authors in the title, introduction, and discussion. If I understand the goal of the paper, the authors want to speed up the process for vaccine development; however, there is no discussion of the time savings or any other metric that demonstrates how the new technology is an improvement. Perhaps the authors could make a few comparisons between the process detailed within the manuscript and current vaccine production timelines, resources, etc? Such a comparison would increase the impact of this manuscript and give the reader a clear understanding of why this technology is an improvement over the existing.
Below are line comments:
Line 94: “[for review…” seems to be an editing oversight. Please correct this reference.
Line 98: The authors have noted twice now that MVA is used to rapidly respond and manufacture vaccines but are also saying that the virus in incapable of replication; while the authors also mention that this phenomenon occurs in mammalian cells, it is unclear to the reader how the MVA is manufactured to adequate national response. Please address this oversight in manufacture to provide a clear case for the MVA vector as candidate for rapid response.
Line 107-110: Just my opinion, but this section of the intro may hugely benefit from a quick discussion of the differences in manufacture, as well as a quick line about timescales for each. Like the comment above from line 98, there’s a lack of information about what this manufacture is and I think it would set up your discussion appropriately. The next paragraph regarding “more swiftly…” making these vaccines would benefit from a discussion of the timeframes.
Line 166-167: Any sequences specific to this assay should be detailed here.
Discussion Line 502-522: It’s great that you’re drilling into the rationale for the paper but I’m curious if you could take it a step further. How much does this save you (time, cost, safety, etc.)? How does this streamline steps from manufacture to shots in arms?
Author Response
Reviewer 3:
- The manuscript titled, “Rapid development of Modified Vaccinia virus Ankara (MVA)-based vaccine candidates against Marburg virus suitable for clinical use in humans,” details the authors efforts to quicken the development process of MVA vectored vaccines for Marburg virus, and perhaps other viral threats. It is a well written manuscript that accurately documents the research conducted to insert tools in the vector background that will help streamline vector production. The authors use Marburg as their test candidate and show relevant immune responses in mice after vaccination compared to MVA. The only critique I can offer is that there is a general lack of impact regarding the purpose laid out by the authors in the title, introduction, and discussion. If I understand the goal of the paper, the authors want to speed up the process for vaccine development; however, there is no discussion of the time savings or any other metric that demonstrates how the new technology is an improvement. Perhaps the authors could make a few comparisons between the process detailed within the manuscript and current vaccine production timelines, resources, etc? Such a comparison would increase the impact of this manuscript and give the reader a clear understanding of why this technology is an improvement over the existing.
A: We greatly appreciate the reviewer´s overall positive response and we thank the reviewer for this helpful suggestion. The two backbone viruses, MVA-GFP and MVA-mCherry, allowed us to speed up the isolation of clonal isolates of newly generated recombinant MVA viruses to be used as vaccines. In addition, the generation of the two backbone viruses under pre-GMP conditions allows us to generate recombinant MVA candidate vaccines that meet the requirements of vaccine manufacturing for humans. Thus, the amplified and characterized virus can be directly shipped to potential vaccine producers after successful in vitro and in vivo characterization. We added the impact of our improved technology in the introduction and discussion where applicable (lines 91-94, 111-115, 124-126, 591-599).
- Line 94: “[for review…” seems to be an editing oversight. Please correct this reference.
A: Thank you for the hint. We clarified the references.
- Line 98: The authors have noted twice now that MVA is used to rapidly respond and manufacture vaccines but are also saying that the virus in incapable of replication; while the authors also mention that this phenomenon occurs in mammalian cells, it is unclear to the reader how the MVA is manufactured to adequate national response. Please address this oversight in manufacture to provide a clear case for the MVA vector as candidate for rapid response.
A: Thank you for the useful suggestion. MVA can infect mammalian cells upon vaccination in animals or humans and initiates viral gene expression to start the replication process. However, viral replication is blocked later during infection, and no infectious particles are assembled and released from the infected cells. This is a key feature of recombinant MVAs and is responsible for their high safety profile as vaccine. However, since MVA still possesses full replication capacity in avian cells, avian cells are used to produce MVA vaccines. To produce MVA at an industrial scale, monolayers of cells of avian origin are infected, resulting in cell culture preparations with high viral load. These cell culture preparations are purified to obtain vaccine preparations. However, our newly established system for the generation of recombinant MVAs has no impact on the manufacturing, but we are able to speed up the workflow in the laboratory for a rapid generation and conduction of quality controls of candidate vaccines. We could halve the time to obtain a positive clonal isolate when using fluorescent MVA backbone viruses compared to non- fluorescent parental MVA viruses. We provided information about the manufacturing process of MVA for rapid response in the manuscript at lines 111-115, 124-132, 579-586, 591-599.
- Line 107-110: Just my opinion, but this section of the intro may hugely benefit from a quick discussion of the differences in manufacture, as well as a quick line about timescales for each. Like the comment above from line 98, there’s a lack of information about what this manufacture is and I think it would set up your discussion appropriately. The next paragraph regarding “more swiftly…” making these vaccines would benefit from a discussion of the timeframes.
A: Thank you for the very helpful suggestion. We added information about the manufacturing process (see lines 111-116, 125-132).
- Line 166-167: Any sequences specific to this assay should be detailed here.
A: Thank you for the hint. We added detailed information about sequence specificities (see lines 200-220).
- Discussion Line 502-522: It’s great that you’re drilling into the rationale for the paper but I’m curious if you could take it a step further. How much does this save you (time, cost, safety, etc.)? How does this streamline steps from manufacture to shots in arms?
A: Thank you for the useful suggestion. We added information about the benefit of using the two MVA backbone viruses for rapid vaccine development (see lines 591-599).

Reviewer 4 Report
Comments and Suggestions for Authors
The paper "Rapid Development of Modified Vaccinia Virus Ankara (MVA)-based Vaccine Candidates Against Marburg Virus Suitable for Clinical Use in Humans" describes a project to create and characterise recombinant MVA vaccines expressing MARV glycoprotein (GP) and nucleoprotein (NP) as potential candidates for rapid response to MARV outbreaks. Marburg virus disease (MVD), a severe and frequently fatal hemorrhagic fever, has spread rapidly throughout both endemic and non-endemic areas, emphasising the critical need for effective vaccinations. While the authors present an overview of MARV virology and propose MVA-based vaccines as a remedy, some areas of the text need to be clarified and strengthened in order to improve its scientific rigour. The methodology, while usually comprehensive, lacks appropriate context for individual experimental decisions, such as dosage and animal model explanations, which might be reinforced with evidence from previous investigations. Furthermore, statistical analyses are limited, and comments about data variability and limits are inadequate, reducing the overall interpretability and robustness of the study's conclusions.
Major Comments
- Lines 24-29: The summary briefly highlights how MARV has expanded to new regions of Africa, emphasising the necessity for vaccinations. However, it does not emphasise the originality or specific benefits of the MVA-based vaccine over current MARV vaccines. Clarifying how this candidate differs in predicted efficacy, speed, or safety would enhance the abstract's impact.
- Lines 46-68: The introduction emphasises the severity and epidemiology of MARV, but fails to place the current investigation into the greater context of MARV vaccine development. It would be useful to compare current techniques, such as ChAd3 and mRNA-based platforms, with the benefits of MVA-based vaccinations.
- Lines 78–84: This section briefly contrasts MVA and mRNA vaccines, using COVID-19 as an example. It would be useful to explain why an MVA vector is predicted to deliver longer-lasting protection or higher stability than mRNA in MARV vaccines.
- Lines 112-119: The use of fluorescent markers (e.g., GFP, mCherry) for recombination verification is new, but the text lacks precise quality control checks to ensure recombination purity. Adding steps to verify purity (e.g., quantification of fluorescent cells, numerous rounds of screening) would improve methodological transparency and reproducibility.
- Lines 189–195: Ethical compliance for animal usage is stated, but no information on mouse group size calculations, animal handling standards, or exclusion criteria for adverse reactions is provided. This absence reduces the rigour of animal model testing while failing to address potential causes of experimental bias.
- Lines 197-202: The vaccine dose (10^7 PFU) and route of administration (intramuscular) are specified, but no reason is provided. A rationale based on past investigations or preliminary tests would supplement the scientific reasoning for these decisions.
- Lines 249-275: This part covers the creation and characterisation of MVA-mCherry and MVA-GFP viruses, but lacks information on genetic stability verification methods. Although PCR analysis is indicated, it would be helpful to know whether sequencing or other methods were employed to ensure steady insertion and expression of target genes.
- Lines 310-327: Immunofluorescence labelling verified MARV-GP expression in Vero cells, but no quantification of expression levels was provided (e.g., signal intensity comparison). Quantitative data would provide more evidence for stable protein expression, increasing the reliability of the findings.
- Lines 406-411: ELISA data are reported as means, but the range or variance between samples is not stated. This is necessary to analyse inter-individual response variability. Including statistical information (e.g., standard deviation) and specific p-values would improve the understanding of vaccine-induced immune responses.
- Lines 480-484: The discussion briefly references MARV's growth in Africa, but does not explicitly relate how this MVA-based vaccine candidate could precisely address issues posed by MARV outbreaks in these settings (e.g., temperature stability and rapid deployment). This would make the study's findings more relevant to public health goals.
- Lines 501-503: The vaccine's claim of rapid availability lacks quantitative data on preparation and reaction time. This assertion might be supported by providing particular timings (for example, reducing the number of passages) or referencing timelines from previous studies.
- Lines 553-555: The authors discuss T-cell responses to MARV-GP, highlighting its multifunctionality. However, without a comparison to benchmarks (such as T-cell responses in naturally infected people or other vaccinations), it is difficult to determine the importance of this reaction. Including comparable data or control benchmarks would provide a more accurate understanding of these results.
- Lines 594-596 mention the success of the MVA-EBOV-NP vaccine in protecting mice from Ebola. However, direct comparisons or bridging information on MARV and Ebola immunology (e.g., cross-reactivity or structural similarities) would help readers understand how lessons from Ebola studies apply to MARV.
Minor Comments
- The manuscript lacks detailed rationale for methodological choices such as dosage, marker selection, and observation time. Including precise, data-driven rationales would improve the manuscript's methodological rigour.
- Statistical information are limited, particularly in the immune response data. An improved presentation of statistical analysis (e.g., confidence intervals, p-values) would boost credibility. Furthermore, the study fails to address potential limits and biases, such as variability in animal response or translation challenges to human models.
Comments on the Quality of English Language
The English could be improved to more clearly express the research.
Author Response
Reviewer 4:
- The paper "Rapid Development of Modified Vaccinia Virus Ankara (MVA)-based Vaccine Candidates Against Marburg Virus Suitable for Clinical Use in Humans" describes a project to create and characterise recombinant MVA vaccines expressing MARV glycoprotein (GP) and nucleoprotein (NP) as potential candidates for rapid response to MARV outbreaks. Marburg virus disease (MVD), a severe and frequently fatal hemorrhagic fever, has spread rapidly throughout both endemic and non-endemic areas, emphasising the critical need for effective vaccinations. While the authors present an overview of MARV virology and propose MVA-based vaccines as a remedy, some areas of the text need to be clarified and strengthened in order to improve its scientific rigour. The methodology, while usually comprehensive, lacks appropriate context for individual experimental decisions, such as dosage and animal model explanations, which might be reinforced with evidence from previous investigations. Furthermore, statistical analyses are limited, and comments about data variability and limits are inadequate, reducing the overall interpretability and robustness of the study's conclusions.
A: We thank the reviewer for the overall positive feedback. We added information on the dosage decision and the animal model where applicable (lines 249-252, 560-567). In addition, we updated the phrasing on statistical analysis where applicable (lines 302-303, 467-468, 473-475, 508-510, 547-549).
- Lines 24-29: The summary briefly highlights how MARV has expanded to new regions of Africa, emphasising the necessity for vaccinations. However, it does not emphasise the originality or specific benefits of the MVA-based vaccine over current MARV vaccines. Clarifying how this candidate differs in predicted efficacy, speed, or safety would enhance the abstract's impact.
A: We agree with the reviewer, and we included an additional section to clarify the specific benefits of MVA based vaccines (see lines 32-35).
- Lines 46-68: The introduction emphasises the severity and epidemiology of MARV, but fails to place the current investigation into the greater context of MARV vaccine development. It would be useful to compare current techniques, such as ChAd3 and mRNA-based platforms, with the benefits of MVA-based vaccinations.
A: We agree with the reviewer, and we included an additional section to compare MVA vector to ChAd3 and mRNA-based platforms (see lines 72-75).
- Lines 78–84: This section briefly contrasts MVA and mRNA vaccines, using COVID-19 as an example. It would be useful to explain why an MVA vector is predicted to deliver longer-lasting protection or higher stability than mRNA in MARV vaccines.
A: We agree with the reviewer, and we included an additional section to explain why MVA vector is an appropriate addition to mRNA-based vaccines (see lines 91-94).
- Lines 112-119: The use of fluorescent markers (e.g., GFP, mCherry) for recombination verification is new, but the text lacks precise quality control checks to ensure recombination purity. Adding steps to verify purity (e.g., quantification of fluorescent cells, numerous rounds of screening) would improve methodological transparency and reproducibility.
A: Thank you for the suggestion. The two MVA backbone viruses were analyzed in depth according to our standardized quality control procedures. This included confirmation of genetic stability by using a PCR reaction targeting the 6 MVA-specific deletion sites. In addition, we used PCR assays targeting the genomic sites that has been used to insert the MARV-antigens and the C7L gene. These PCR-specific quality controls were performed after plaque purification and virus amplification. In addition, stable and correct protein expression was confirmed by western blot analysis and fluorescence microscopy in 4 different cell lines (A549, CEF, HeLa, DF-1). In addition, full replication capacity in the avian CEF cells and replication deficiency in three human cell lines (HaCat, A549, HeLa) was confirmed. MVA as a poxvirus is very stable due to its nature as a DNA virus and therefore, mutations in the genome during the purification and amplification procedure are neglectable. The isolation of a positive clonal isolate included 5-8 rounds, and afterward, the virus was slowly amplified at a low MOI (0.05) by consecutive rounds of infections for plaque purification. Therefore, cell cultures in increasing numbers were infected, and the quality control procedures as described were performed, confirming the stability of the virus. These results are included in Figures S1, S2 and S3.
- Lines 189–195: Ethical compliance for animal usage is stated, but no information on mouse group size calculations, animal handling standards, or exclusion criteria for adverse reactions is provided. This absence reduces the rigour of animal model testing while failing to address potential causes of experimental bias.
A: Thank you for the hint. We added the information to the Material & Method section (see lines 237-240, 252-256).
- Lines 197-202: The vaccine dose (10^7 PFU) and route of administration (intramuscular) are specified, but no reason is provided. A rationale based on past investigations or preliminary tests would supplement the scientific reasoning for these decisions.
A: Thank you for the suggestion. The i.m. application has been chosen because it is the preferred route of administration for MVA-based vaccines in humans in clinical studies. In previous studies with mice, we observed that a dose of 107 PFU was well tolerated and highly immunogenic. Due to that, we decided to use 107 PFU for the current study. We added the information in the Material & Method section (see lines 249-252).
- Lines 249-275: This part covers the creation and characterisation of MVA-mCherry and MVA-GFP viruses, but lacks information on genetic stability verification methods. Although PCR analysis is indicated, it would be helpful to know whether sequencing or other methods were employed to ensure steady insertion and expression of target genes.
A: Thank you for the suggestion. At this stage of preclinical development, we have not performed genome sequencing. As mentioned above, we performed western blotting, fluorescence microscopy, and multiple-step growth kinetics as quality controls. These standardized methods are accepted to confirm the genetic stability and stable expression of the foreign antigens of our MVA candidate vaccines for production at industrial scale.
- Lines 310-327: Immunofluorescence labelling verified MARV-GP expression in Vero cells, but no quantification of expression levels was provided (e.g., signal intensity comparison). Quantitative data would provide more evidence for stable protein expression, increasing the reliability of the findings.
A: Thank you for the useful thoughts. We did not quantify the expression level, as we infected the cells at a low MOI to demonstrate the intracellular location of the expressed proteins (MARV-GP, MARV-NP) on a single-cell level which is sufficient to correlate the activation of MARV-specific immune responses as induced in vivo in the mice with the expression of MARV-antigens by MVA.
- Lines 406-411: ELISA data are reported as means, but the range or variance between samples is not stated. This is necessary to analyse inter-individual response variability. Including statistical information (e.g., standard deviation) and specific p-values would improve the understanding of vaccine-induced immune responses.
A: Thank you for the helpful suggestion. We added more detailed information about analyzing the ELISA data where applicable (see 467-468, 473-475,).
- Lines 480-484: The discussion briefly references MARV's growth in Africa, but does not explicitly relate how this MVA-based vaccine candidate could precisely address issues posed by MARV outbreaks in these settings (e.g., temperature stability and rapid deployment). This would make the study's findings more relevant to public health goals.
A: This is a very appropriate suggestion. We now include a section about the advantages of MVA-based vaccination strategies against MARV, including their temperature stability, well-approved safety, rapid deployment, and immunogenicity (see lines 579-586-608).
- Lines 501-503: The vaccine's claim of rapid availability lacks quantitative data on preparation and reaction time. This assertion might be supported by providing particular timings (for example, reducing the number of passages) or referencing timelines from previous studies.
A: Thank you for the suggestion. We added information on time savings and reducing the number of plaque passages where applicable (see lines 125-126, 591-599, 616-619,.
- Lines 553-555: The authors discuss T-cell responses to MARV-GP, highlighting its multifunctionality. However, without a comparison to benchmarks (such as T-cell responses in naturally infected people or other vaccinations), it is difficult to determine the importance of this reaction. Including comparable data or control benchmarks would provide a more accurate understanding of these results.
A: This was a good suggestion and we discussed T cell responses in MARV survivors (see lines 655-659,).
- Lines 594-596 mention the success of the MVA-EBOV-NP vaccine in protecting mice from Ebola. However, direct comparisons or bridging information on MARV and Ebola immunology (e.g., cross-reactivity or structural similarities) would help readers understand how lessons from Ebola studies apply to MARV.
A: This is a very appropriate suggestion and we agree with the reviewer that bridging information on MARV and Ebola immunology would significantly support the conclusion. For this, we added a section within this part of the discussion on immunological cross-reactivity between MARV and Ebola (see lines 679-683).
- The manuscript lacks detailed rationale for methodological choices such as dosage, marker selection, and observation time. Including precise, data-driven rationales would improve the manuscript's methodological rigour.
A: Thank you for the helpful suggestion. If applicable, we added the rationality of methodological choices. Overall, generation, in vitro, and in vivo characterization of the MVA backbone viruses and the MVA vaccine candidates were performed according to protocols that have been well-established in our lab for years and are accepted by local competent authorities and manufacturing companies. This includes the choice of time points to collect cell cultures (e.g. for western blotting or immunofluorescence staining), the choice of cell lines to confirm replicative capacity or replication deficiency, or the choice of the insertion site(s) for the target antigens. For the in vivo studies, the immunization schedule (21-day interval), the dosage, and the route of application have been proven to be well tolerated and highly immunogenic in previous studies. The scope of this study was to confirm the suitability of our backbone viruses to rapidly generate and characterize candidate vaccines against (re-) emerging viruses with MARV as an example. A detailed analysis including efficacy testing or different doses of the vaccines will be conducted in a next study.
- Statistical information are limited, particularly in the immune response data. An improved presentation of statistical analysis (e.g., confidence intervals, p-values) would boost credibility. Furthermore, the study fails to address potential limits and biases, such as variability in animal response or translation challenges to human models.
A: This was a good suggestion and we included statistical information were applicable and added potential limits of our study. In addition, we also included a section describing the limitations and biases of the study (see lines 717-722).

Reviewer 5 Report
Comments and Suggestions for Authors
This is an excellent and comprehensive manuscript addressing an important unmet need. I have no substantive criticisms.
Author Response
-
Reviewer 5:
- This is an excellent and comprehensive manuscript addressing an important unmet need. I have no substantive criticisms.
A: We greatly appreciate the reviewer´s overall positive response.

Round 2
Reviewer 1 Report
Comments and Suggestions for Authors
The authors have addressed most of my questions and comments. I only have a few comments left.
-Figure 1d, 1f. It is better not to use the same name (MBA) for different samples.
-Сut off for ELISA. From the authors' answer, I got the impression that they use not the mean+4SD but the absorbance value of 0.2 as a cutoff. If so, this should be clarified in the materials and methods section. The mean value + 4SD at the dilution 1/100 can be any and is not necessarily greater than or equal to 0.2.
-Wherever points of individual measurements are not presented in the graph (box plots), the number of measurements should be clarified. This is critical when the error bars are represented by the SEM. This refers to both the MS and Supplementary files.
-In the captions to the figures in the Supplementary files, it is also necessary to indicate what the error bars represent.
-line 674. “Orthomyxoviruses”. This word is better written with a lowercase letter.
Author Response
Revision Note
We thank the reviewer for the thoughtful comments and constructive suggestions, which enabled us to resubmit a clearly improved manuscript. In the following, we listed all points raised by the reviewer, our response to them, and the actions we took to advance our work. In the marked version, we highlighted all changes in the manuscript in yellow.
Reviewer 1:
-Figure 1d, 1f. It is better not to use the same name (MBA) for different samples.
Thank you for the hint. We assume that this refers to the MVA sample included in Figures 1d and 1f. However, in both these assays, DNA of the same empty MVA vector virus is used as a control to confirm the insertion of either the GFP-encoding sequences within the deletion site III or the mCherry-encoding sequences within the I8R-G1L site. Since the PCR assays targeted different sites, different primer pairs were used for Figure 1d and 1f. Thus, we left the same name,” MVA,” for Figure 1d and Figure 1f.
-Сut off for ELISA. From the authors' answer, I got the impression that they use not the mean+4SD but the absorbance value of 0.2 as a cutoff. If so, this should be clarified in the materials and methods section. The mean value + 4SD at the dilution 1/100 can be any and is not necessarily greater than or equal to 0.2.
Thank you for the hint. We apologize for our misleading writing and have changed the ELISA section of the materials and methods accordingly to clarify this.
-Wherever points of individual measurements are not presented in the graph (box plots), the number of measurements should be clarified. This is critical when the error bars are represented by the SEM. This refers to both the MS and Supplementary files.
Thank you for the suggestion. We checked the figure legends, which contain graphs showing mean + SEM, and they all state the n values in the first sentence describing the mice.
-In the captions to the figures in the Supplementary files, it is also necessary to indicate what the error bars represent.
We changed the figures accordingly.
-line 674. “Orthomyxoviruses”. This word is better written with a lowercase letter.
We changed it accordingly.
